# Deep Learning-Based Multimode Fiber Distributed Temperature Sensing

**DOI:** 10.3390/s25092811

**Published:** 2025-04-29

**Authors:** Luxuan Yang, Xiaoyan Wang, Tong Wu, Huichuan Lin, Songjie Luo, Ziyang Chen, Yongxin Liu, Jixiong Pu

**Affiliations:** 1Fujian Provincial Key Laboratory of Light Propagation and Transformation, College of Information Science & Engineering, Huaqiao University, Xiamen 361021, China; 23013082014@stu.hqu.edu.cn (L.Y.); xiaoyan_wang_3@foxmail.com (X.W.); 22013082013@stu.hqu.edu.cn (T.W.); songjie@hqu.edu.cn (S.L.); ziyang@hqu.edu.cn (Z.C.); 2College of Physics and Information Engineering, Minnan Normal University, Zhangzhou 363000, China; lhc1810@mnnu.edu.cn; 3National Laboratory on High Power Laser and Physics, Shanghai Institute of Optics and Fine Mechanics, Chinese Academy of Sciences, Shanghai 201800, China

**Keywords:** convolutional neural networks, multimode fibers, temperature prediction, position prediction, speckle imaging, distributed sensing

## Abstract

**Highlights:**

**What are the main findings?**

**What are the implications of the main findings?**

**Abstract:**

As a laser beam passes through a multimode fiber (MMF), a speckle pattern is generated, which is sensitive to temperature, thereby making the MMF a temperature-sensing element. A deep learning technique is employed to the MMF-based temperature sensor, to obtain high-precision temperature sensing. We designed an MMF-based temperature-sensing configuration and developed a dual-output Convolutional Neural Network (CNN) for predicting both the temperature and the position of the heating point, and we constructed a dataset. It was shown that the location prediction accuracy reached 100%, while the temperature prediction accuracy (within a ±1 °C error margin) was 100% and 95.12% in the two experiments, respectively. The precision of the predicting heating point was less than 1 cm. Different types of MMFs were used in temperature measurements, showing that the accuracy remained quite high. This non-contact, high-precision MMF-based temperature measurement method, driven by deep learning, is suitable for applications in hazardous environments.

## 1. Introduction

The rapid advancement in multimode fiber (MMF) sensing has made fiber-optic monitoring a key research area in industries, healthcare, and environmental monitoring [1,2,3,4,5], with applications in bend detection [6], temperature monitoring [7], fluid dynamics [8], corrosion monitoring [9], fire warning [10], and weak magnetic field detection [11]. Traditional fiber-optic temperature sensing, which relies on the fiber’s reflectivity or temperature-dependent properties [12,13], faces challenges such as high costs, complex installation, and limited accuracy, hindering its widespread use.

In recent years, the combination of MMFs and deep learning has gained increasing attention. For example, Convolutional Neural Network (CNN) techniques have been used to classify the intensity of speckle patterns in MMFs [1], achieve the high-fidelity imaging of bent MMFs [14], and learn the input–output relationship in 0.75 m MMFs [15]. A method combining Short-Time Fourier Transform and Resnet152 has been proposed to identify various sensing events, such as climbing, collision, and cutting [16]. Real-time strain distribution interpretation for detecting spatial cracks has also been realized using deep learning [17].

In temperature measurement, CNNs combined with Raman Optical Time Domain Reflectometry signals have been used for earth and rockfill dam monitoring [7]. Distributed temperature sensing via Raman scattering can measure temperature profiles [18], but it requires understanding complex physical models like backscattering theory. Other studies have explored micro-displacement and dimension measurements using Fiber Bragg Grating cascading [19]. Alternatively, using deep learning to predict temperatures from speckle patterns bypasses complex physics, simplifying the process.

In speckle pattern analysis, laser beams passing through MMFs create speckle patterns due to multiple reflections and interferences. These patterns, characterized by random bright and dark spots, change with external conditions such as temperature. Deep learning has been used to enhance speckle pattern sensing [20], suppress noise [21], enable multiplexed sensing [22], achieve a 0.01 mm displacement prediction [23], and develop a wavelength meter by correlating speckle patterns with the light source wavelength [24]. It has also been applied to detect the curvature and position of bent MMFs [25,26] and monitor respiratory rates through dynamic speckle videos [27]. These studies highlight the potential of speckle patterns in various sensing applications but have not yet addressed temperature distributed sensing.

This study innovatively combines MMF with deep learning to achieve distributed temperature sensing via speckle image analysis, overcoming traditional limitations and offering a new direction for fiber-optic sensing. By using CNNs, we analyze speckle images from MMFs to predict the temperature and spatial position at different heating points. Two types of MMFs with different specifications were tested under various temperature and heating conditions. The experiments used 65 cm and 100 cm bare MMFs with 400 μm and 600 μm cladding diameters, covering a temperature range from 20 °C to 99 °C with a 1 °C accuracy. Speckle images collected from heated MMF points were used to build a dataset, which was then processed by the CNN model for the dual-task prediction of temperature and position.

## 2. Principle

### 2.1. Structure of Fiber

An optical fiber typically consists of a core, cladding, coating, buffer layer, and jacket. The core is the central part of the fiber, responsible for transmitting optical signals. Light propagates within the core through the principle of total internal reflection. The cladding, which surrounds the core, has a slightly lower refractive index than the core and is designed to confine the optical signal within the core, preventing light leakage. The coating, located outside the cladding, protects the fiber from mechanical damage, moisture ingress, and chemical corrosion. It also provides flexibility, facilitating the bending and installation of the fiber. The buffer layer further protects the fiber from external mechanical stress and environmental factors, offering additional flexibility and tensile strength. The jacket, as the outermost protective layer, provides extra mechanical protection to prevent damage during installation and use, as well as protection against chemical corrosion and environmental influences.

In this experiment, a commercially available bare multimode optical fiber with a step-index profile was used. The fiber structure comprised only the core, cladding, and coating, without a buffer layer or jacket. This simplified structure is advantageous for sensing temperature changes. The fiber featured a pure silica glass core, a fluorine-doped silica glass cladding, and a polyimide resin coating, which could withstand a temperature range from −65 °C to 300 °C. Two fibers were employed in the experiment, with core diameters of 600 μm and 400 μm, respectively. After cladding, the outer diameters were approximately 660 μm and 440 μm (with an error range of ±5 μm). Following the application of the coating, the total outer diameters were 960 μm and 700 μm (with an error range of ±20 μm). This fiber operated within a wavelength range of 400 nm to 2500 nm.

### 2.2. Principle of MMF Speckle Pattern Sensor

The multimode optical fiber distributed sensor investigated in this study, also known as the optical fiber speckle pattern sensor, is a modality-based sensor. It operates on the principle of modal interference among all guided modes within the fiber, resulting in distinct output intensity distributions. This sensor achieves high-sensitivity sensing by exploiting the sensitivity of speckle patterns to external physical quantities such as temperature, strain, and vibration. The formation of speckle patterns is closely related to the coherence of the light source [28,29]. When a coherent light source (such as a laser) is used, light waves of different modes share the same frequency and phase relationship, leading to more pronounced interference and the formation of granular speckle patterns with significant intensity variations. In contrast, when an incoherent light source is employed, the phase relationships of light waves in different modes vary randomly, resulting in less noticeable interference and a more uniform speckle pattern. Therefore, a 633 nm helium–neon laser was selected as the light source in this experiment.

### 2.3. Ray Theory and Modal Theory

Multimode optical fibers can guide many modes with different phase velocities, and the interference of these modes can be explained using ray theory and mode theory.

Ray theory is based on geometrical optics, treating light waves as rays propagating within the fiber. In multimode fibers, rays enter the fiber at different angles and undergo multiple reflections at the core–cladding interface. Rays propagating along the fiber axis correspond to the lowest-order mode, with the shortest propagation path and the shortest time to reach the end. In contrast, rays with high-angle reflections correspond to higher-order modes, with longer propagation paths and longer times to reach the end. Due to the different propagation path lengths of various modes, they arrive at the end at different times. This time delay causes the light waves of different modes to interfere at the output end, creating an intensity distribution with alternating bright and dark regions, known as the speckle pattern.

Mode theory is based on the propagation characteristics of electromagnetic waves in the fiber. Modes in the fiber are electromagnetic field distributions that satisfy Maxwell’s equations and boundary conditions, with each mode having a different phase velocity and spatial distribution. When multiple modes propagate simultaneously in the fiber, they interfere at the output end. Since different modes have different phase velocities and spatial distributions, their phase relationships change with propagation distance and time, leading to variations in the output light field’s intensity distribution. These variations are the source of the speckle pattern [30].

The number of modes depends on the fiber’s geometrical shape (such as its core diameter) and the frequency of the light. The normalized frequency V is a key parameter used to calculate the number of modes that can propagate in the fiber, as shown in Equation (1):(1)$$V=\frac{2\pi a}{\lambda }\sqrt{n_{1}^{2}-n_{2}^{2}}$$

Here, $n_{1}$ is the core refractive index, $n_{2}$ is the cladding refractive index, a is the core diameter, and λ is the wavelength of light. The numerical aperture (NA) of the fiber is given by Equation (2):(2)$$NA=\sqrt{n_{1}^{2}-n_{2}^{2}}$$

Thus, the normalized frequency V can also be expressed as Equation (3):(3)$$V=\frac{2\pi a}{\lambda }NA$$

Based on the normalized frequency V, the number of modes M supported by the fiber can be approximated as follows. For step-index fibers, the mode number is calculated using Equation (4):(4)$$M\approx \frac{V^{2}}{2}$$

For graded-index fibers, the mode number is calculated using Equation (5):(5)$$M\approx \frac{V^{2}}{4}$$

In this experiment, two sets of step-index fibers were used, with core diameters of 600 μm and 400 μm, respectively. The laser wavelength was 633 nm, and the numerical aperture (NA) was 0.22 for both fibers. According to Equation (3), the normalized frequencies of these two sets of fibers were calculated to be 1310 and 873, respectively. Using Equation (4), the mode numbers were found to be 858,050 and 381,065, respectively.

It can be seen from Equation (3) that a larger core diameter results in a higher normalized frequency, and consequently, a greater number of modes. Since the number of speckles is approximately equal to the number of modes, a higher mode number leads to more complex speckle patterns at the output, providing richer information. Additionally, a larger numerical aperture (NA) and a shorter wavelength of light will also increase the normalized frequency, thereby increasing the mode number and the number of speckles. In this experiment, the numerical aperture and the wavelength of light were kept constant, making the core diameter the primary factor affecting the mode number and the number of speckles.

### 2.4. Speckle Patterns and Perturbations

In MMF speckle pattern sensors, the distribution of speckles varies with perturbations, while the total intensity of the modes remains constant. Perturbations cause changes in the refractive index of the fiber, which in turn induce small phase shifts that affect the speckle pattern. The speckle pattern appearing at the output end of the fiber changes due to phase variations within the multimode fiber. As the distribution of speckles changes, the modes within the fiber are redistributed, resulting in intensity variations. The effect of perturbations on the fiber can be described by the following equations [30]. Before perturbation, the light intensity $I_{0}$ can be expressed as Equation (6):(6)$$I_{0}\left(x,y\right)=\left|\sum_{m=1}^{M}\sum_{n=1}^{M}[a_{0m}\left(x,y\right)a_{0n}\left(x,y\right)]\times e^{\left(j\emptyset_{0m}\left(x,y\right)-\emptyset_{0n}\left(x,y\right)\right)}\right|$$

Here, N is the total number of pixels, M is the number of modes, and $a_{0m}\left(x,y\right)$ and $\emptyset_{0m}\left(x,y\right)$ represent the amplitude and phase distributions, respectively. When an external perturbation is applied to the fiber, the speckle pattern changes due to the modification of the modes. If the change in the m-th mode is $\delta_{am}$ in amplitude and $\delta \emptyset_{m}$ in phase, the corrected light intensity after perturbation [30] is given by Equation (7):(7)$$I\left(x,y\right)=\left|\sum_{m=1}^{M}\sum_{n=1}^{M}[a_{0m}\left(x,y\right)+{\delta_{am}][a}_{0n}+\delta_{an}]\times e^{\left(j{\{\emptyset }_{0m}\left(x,y\right)-\emptyset_{0n}\left(x,y\right)+\delta \emptyset_{m}-\delta \emptyset_{n}\right)}\right|$$

Due to the dependence of the output modes on external disturbances, MMF speckle pattern sensors can be used for high-sensitivity and low-cost sensing applications. By analyzing the changes in speckle patterns using image processing techniques, the causes of perturbations can be identified. For example, temperature changes can cause the thermal expansion and contraction of the fiber, altering the paths of light within it. By analyzing the spatial intensity with and without perturbations using Equations (6) and (7), and combining this with statistical features used for image analysis, an optical fiber speckle pattern sensor (FSS) can be designed. In this paper, deep learning is employed to analyze changes in speckle patterns and correlate them with heating locations and temperatures, thereby achieving the high-precision detection of temperature distributions.

## 3. Methods

### 3.1. Experimental Setup

In this experiment, a helium–neon laser served as the light source, coupled into a bare fiber coated with polyimide resin via a mirror and objective lens. The polyimide coating ensures the fiber’s resistance to high-temperature environments. Non-heated fiber sections were thermally isolated using FR-4 fiberglass boards, while brass blocks (8 mm wide) were placed at both fiber ends to clamp and heat the designated sampling points, ensuring rapid and uniform heating. The heating device employed in this study is a DB-1-type digital-display stainless steel electric hot plate. This device facilitates intuitive temperature monitoring through its digital display and ensures rapid and uniform heating. Its temperature control range is from room temperature to 300 °C, with a precision of 1 °C, and the heating power is 500 W.

During the experiment, it was observed that slight fiber movements, bending, personnel movements, air conditioning airflow, and vibrations caused by objects placed on the experimental table—all these environmental factors—led to continuous changes in the speckle pattern. Such changes, especially in the early stages of the experiment, could easily be mistaken for dynamic signals, thereby interfering with the data analysis. This indicates that the sensing system is highly sensitive to external mechanical influences such as tension, stress, torsion, and vibration. However, after employing fiber clamps, the stability of the speckle pattern was significantly enhanced, and external mechanical influences were effectively suppressed. Therefore, we used two clamps, C1 and C2, to secure the fiber, thereby ensuring the system could reliably estimate the temperature and locate the heating points. Speckle images were captured using an industrial CCD camera equipped with a neutral-density filter to protect against an excessive light intensity. The exposure time was set to 8000 us to enhance the speckle clarity and contrast. The experimental setup is shown in Figure 1.

In the experiment, a 633 nm He-Ne laser was employed as the light source. Two plane mirrors, designated as M1 and M2, were used to direct the laser beam. An objective lens (OBJ) with a magnification of 20 and a numerical aperture (NA) of 0.4 was utilized for focusing purposes. Fiber clips (C1 and C2) were employed to secure the optical fibers in place. A neutral-density filter (ND) was used to attenuate the laser intensity when necessary. The camera used in this study is the GT2050 model with a GigE interface, manufactured by Allied Vision. This camera features a resolution of 2048 × 2048 pixels, with a pixel depth of 8 bits. The image data format is Mono8, and the pixel size is 5.5 μm.

The experiment was divided into two groups. In the first group, a bare fiber with a core diameter of 600 μm and a total length of 65 cm was used. A heating point (B) was selected at the midpoint of the fiber, and additional heating points were set at 5 cm to the left of B (point A) and 6 cm to the right of B (point C). In the second group, a bare fiber with a core diameter of 400 μm and a total length of 100 cm was employed. Sampling points included D, E, F, G, and H, with intervals of 1 cm, 2 cm, 4 cm, and 5 cm, respectively.

### 3.2. Construction of Temperature and Position Datasets

In the experiment, speckle images were captured for each sampling point. For instance, at point A, 10 speckle images were first captured continuously at room temperature (20 °C). Subsequently, another 10 images were captured at 21 °C, and this process was repeated at 1 °C intervals up to 99 °C. A total of 10 speckle images were captured at each temperature, resulting in 80 temperature points and a total of 800 speckle images. Similarly, speckle images were captured at points B, C, D, E, F, G, and H within the temperature range of 20 °C to 99 °C.

The maximum temperature tolerance of the insulating board is 180 °C, and its thermal insulation performance deteriorates above 120 °C, affecting the temperature at locations other than the target heating point along the fiber. Therefore, 99 °C was chosen as the upper temperature limit for the current experiment. When heating with the device from room temperature to 50 °C, it took approximately 3 min for the temperature to stabilize at each 1 °C increment. From 50 °C to 99 °C, it took about 2 min for the temperature to stabilize at each 1 °C increment. The higher the temperature, the shorter the time required to reach a steady state. After each temperature stabilized, 10 images were captured and saved as image files, which took approximately 9 s.

The dataset of the first group (fibers with a cladding diameter of 600 µm) includes 2400 speckle images from three points, namely A, B, and C, with the training set containing 1920 speckle images and the test set containing 480 speckle images. Meanwhile, the dataset of the second group (fibers with a cladding diameter of 400 µm) includes 4000 speckle images from five points, namely D, E, F, G, and H, with the training set containing 3200 images and the test set containing 800 speckle images. The dataset structure is illustrated in Figure 2.

To further explore the variability within the datasets, we conducted a similarity analysis on the 10 speckle images acquired at a constant temperature of 20 °C at point B. The results were visualized in the form of a similarity matrix. The similarity metric was based on the Root Mean Squared Error (RMSE) [31], which is defined as follows (8):(8)$$RMSE=\sqrt{\frac{1}{n}\sum_{i=1}^{n}{\left(x_{i}-y_{i}\right)}^{2}}$$

In this context, $x_{i}$ and $y_{i}$ represent the grayscale values of the two images at the i-th pixel, while n denotes the total number of pixels in the image. This formula allows for a quantitative assessment of the differences between the images.

Before the RMSE calculation, the speckle images were normalized. The maximum RMSE value of 1 represents the difference between a pure black and a pure white image. Darker colors in the matrix indicate smaller differences, while lighter colors indicate larger differences.

Figure 3a shows the similarity matrix of images captured at the same temperature. The maximum difference under the same temperature is 0.008, indicating a high degree of similarity between the images captured at the same temperature. We also compared images captured at different temperatures (Figure 3b). Compared with the values of the same temperature, the similarity differences between different temperatures are more significant, and a larger temperature difference leads to a larger similarity difference value. Nevertheless, the maximum value of 0.16 is still relatively small, indicating that the differences among the speckle images in the dataset are not significant.

Although the speckle images may appear visually similar, the differences between the images can still be recognized by a specifically designed algorithm at the pixel level. Specifically, there are slight variations in the grayscale values and texture features of the speckle images, which reflect different temperature or position information. Thus, deep learning methods, particularly Convolutional Neural Networks (CNNs), can be used to learn these subtle differences and provide high-accuracy predictions. The analysis of the similarity matrix further validates the feasibility of deep learning approaches for temperature and position prediction, which holds significant practical application value, especially in optical fiber sensor applications.

### 3.3. Data Preprocessing and CNN Model Design

During the data preprocessing stage, all input images were first resized and normalized. The images, originally captured at a resolution of 2048 × 2048 pixels by the CCD, were reshaped to a size of 256 × 256 pixels to meet the input requirements of the neural network. All the images were resized to a uniform dimension, and their grayscale values were normalized to the range of [0, 1] to enhance the training efficiency and convergence speed of the model.

A deep learning model based on Convolutional Neural Networks (CNNs) was designed in this study to simultaneously predict the temperature and heating location of the optical fiber. The model comprises multiple convolutional layers, pooling layers, batch normalization layers, and fully connected layers. The feature extraction section employs a multi-layer convolutional module, with the number of channels increasing progressively from 1 to 2048, thereby effectively capturing the complex features within the speckle images. The specific architecture of the network model is shown in Figure 4.

The model adopts a dual-task learning framework for separate temperature and position predictions. In this framework, the feature extraction layers are shared, which not only improves the model’s performance on both tasks but also reduces the computational load. Specifically, the temperature prediction task outputs 80 classes (corresponding to 80 temperature levels), and the position prediction task varies depending on the experimental setup. In the first group of experiments, it outputs 3 position categories, and in the second group, it outputs 5 position categories.

In the model, the feature extraction module is composed of several stacked convolutional layers, batch normalization layers, and ReLU activation functions. Each convolutional layer is followed by a batch normalization layer and a ReLU activation function to improve the network’s non-linear representation ability and training stability. The convolution kernel size is 3 × 3, with a stride of 1 and padding of 1 to ensure the feature map’s size remains unaffected by the convolution operation. Pooling is performed using 2 × 2 max pooling (MaxPooling) with a stride of 2 to gradually reduce the feature map’s spatial dimension while expanding the receptive field.

The temperature prediction branch uses global average pooling (AdaptiveAvgPool2d) to reduce the feature map to a size of 1 × 1, which is then flattened into a one-dimensional vector and input into the fully connected layer for classification. The number of output nodes in the output layer corresponds to the number of temperature categories, set to 80 levels for temperature prediction.

The position prediction branch has a similar structure to the temperature prediction branch, performing prediction via global average pooling and fully connected layers. The output node count for this branch is adjusted according to the experimental design: 3 position categories for the first group of experiments, and 5 position categories for the second group. The learning rate for both experimental groups is set to 0.00002, with a batch size of 4.

## 4. Results and Discussion

The results of the two sets of experiments in this study are presented in three steps. First, the prediction of temperature is conducted independently. Second, the prediction of location is performed separately. Finally, the predictions of temperature and location are combined to achieve a simultaneous dual-task prediction.

### 4.1. Results of the First Group of Experiments

When predicting the temperature alone, prediction curves were generated for three points (A, B, and C). To fully assess the model’s performance, two accuracy metrics were used: strict accuracy and Tolerance ±1 °C Accuracy.

Tolerance ±1 °C Accuracy is a metric that allows for a certain range of error (±1 °C) between the model’s predicted values and the true values. As long as the error falls within this range, the prediction is considered correct. The formula for calculating Tolerance ±1 °C Accuracy is given in Equation (9).(9)$$Tolerance\pm 1\;{}^{\circ}C\;Accuracy=\frac{samples\;with\;error\;within\pm 1{}^{\circ}C}{Total\;samples}\times 100\%$$

Strict accuracy refers to the proportion of instances where the model’s predicted values exactly match the true values. Specifically, a prediction is considered correct only when the model’s output is identical to the actual temperature value. This metric reflects the model’s performance in terms of precise prediction, and its calculation formula is given in Equation (10).(10)$$Strict\;Accuracy=\frac{samples\;with\;exact\;match}{Total\;samples}\times 100\%$$

By employing the two types of accuracy described above, we can gain a more comprehensive understanding of the model’s performance under different precision requirements. For the prediction of temperature using the test dataset, a total of 160 speckle images were collected for each heating point. The results shown in Figure 5a–c indicate that the tolerance accuracies for the three points are 100%, 100%, and 100%, respectively, while the strict accuracies are 98.12%, 100%, and 96.25%. These findings demonstrate that the model is capable of accurately predicting temperature variations during the prediction task.

Given the minimal discrepancy between the predicted and set temperatures, and the strong linear correlation between them, we have plotted the corresponding error curves, as shown in Figure 6. The error values were calculated by subtracting the true temperature values from the predicted temperature values. In the figure, the error curve for Heating Point A is represented by orange circular markers. During the prediction process, Heating Point A exhibited prediction errors of 1 °C at three temperature points, namely 48 °C, 72 °C, and 89 °C. The predicted values at these points were each 1 °C lower than the true values, hence the negative values on the error curve. In contrast, the error curve for the heating point denoted by blue square markers indicates zero error, meaning the predicted temperatures were identical to the true temperatures. The green triangular curve, representing another heating point, shows prediction errors of 1 °C at five temperature points: 75 °C, 84 °C, 86 °C, 91 °C, and 98 °C. The predicted values at these points were also each 1 °C lower than the true values.

When predicting the position alone, a confusion matrix was generated based on the prediction results, as shown in Figure 5d. The results indicated that the position prediction accuracy for points A, B, and C was 100%. Specifically, the model accurately predicted the positions of 160 speckle images in the test sets for each of the three locations (A, B, and C). This demonstrates the model’s perfect classification ability in the position prediction task, enabling precise identification of each point’s location.

In the dual-task prediction of temperature and position, the temperature prediction results are shown in Figure 7. The average accuracy of temperature prediction across the three points (A, B, and C) was calculated, revealing a tolerant accuracy of 100% and a strict accuracy of 97.92%. Meanwhile, the position prediction results remained consistent with those in Figure 5d, maintaining an accuracy of 100%.

The results indicate that in practical applications, when a speckle pattern is input, the corresponding temperature and heating location information can be obtained quickly and accurately. To more intuitively demonstrate this effect, as shown in Figure 8, we present six speckle patterns, along with their predicted temperatures and locations, and the actual temperature values at three positions (A, B, and C) are labeled (51 °C and 80 °C, respectively). The results show that the predicted temperatures and locations by the model are in high agreement with the actual values.

### 4.2. Second Group of Experiment Results

In the second group of experiments, the temperature prediction task involved five different points (D, E, F, G, H). By plotting the temperature prediction curves, we obtained the tolerance accuracy and strict accuracy for each point. The specific results are shown in Table 1:

The results indicate that as the fiber length increases, the core diameter decreases, and as the number of modes in the multimode fiber decreases, the model’s ability to extract speckle features diminishes. This leads to a slight decrease in the temperature prediction accuracy compared to the first set of experiments. The reduction in core diameter leads to fewer propagation modes in the fiber, as the number of modes is closely related to the core and cladding diameters, as well as the wavelength of the light. As the number of modes decreases, the extractable speckle features are also reduced, which in turn affects the temperature prediction accuracy. Particularly at point H, the prediction accuracy is lower than at other points. Despite this, the model is still able to maintain a high prediction accuracy across different points, especially within the temperature tolerance range, where the prediction results exhibit a high degree of accuracy.

Similar to the first set of analyses, to more accurately assess the deviation between the predicted temperature values and the true temperature values, an error curve was plotted (as shown in Figure 9). The error values were calculated by subtracting the true temperature values from the predicted temperature values. The results indicate that as the temperature increases, the model’s prediction accuracy gradually declines, with the prediction error beginning to increase significantly. Particularly after the temperature exceeds 65 °C, the maximum prediction error can reach ±7 °C. This may be attributed to the increased difficulty in extracting speckle features as the temperature rises, thereby reducing the prediction accuracy. By comparing Figure 6 and Figure 9, it is evident that the error for the second set of fibers is significantly higher than that for the first set. This increase in error may be related to differences in the type of light and the number of modes in the second set of fibers, which contribute to the relatively lower prediction accuracy of the second set.

For the position prediction alone, a confusion matrix was generated, and the results showed that the position prediction accuracy for points D, E, F, G, and H was 100%. This indicates that the model performed well in the multi-point position prediction task, accurately identifying the location of each point.

In the dual-task prediction of temperature and position, the average accuracy of temperature prediction across the five points (D, E, F, G, and H) was calculated, as shown in Figure 10a. The results revealed a tolerant accuracy of 95.12% and a strict accuracy of 88.12%. However, the deviation in temperature prediction increased after 70 °C. Meanwhile, the confusion matrix for position prediction (Figure 10b) shows that the position prediction accuracy remained at 100% for all points.

Similarly, in practical applications, to provide a more intuitive demonstration, in the second set of experiments, the temperature and location of any speckle pattern in the test set can be accurately predicted by inputting it into the deep learning model. Figure 11 shows six speckle patterns, along with their predicted temperatures and locations. The actual temperature values at five positions (D, E, F, G, and H) are also labeled (51 °C and 80 °C). The results indicate that, except for the prediction of 51 °C at position H, which was estimated as 52 °C (still within a reasonable error range), all other predicted temperatures and locations are consistent with the actual values.

### 4.3. Discussion

In the task of simultaneously predicting temperature and location, both tolerance accuracy and strict accuracy were found to be high, indicating that multitask learning exhibits good synergistic effects in predicting temperature and location. The deep learning model was able to accurately predict temperature and location across different fiber core diameters (400 μm and 600 μm) and fiber lengths (65 cm and 100 cm). Particularly for the 400 μm fiber diameter, the model was capable of accurately distinguishing heating points separated by 1 cm, demonstrating its robustness under various experimental conditions. Within the same fiber and under the condition of the same number of modes, heating points located in the middle position exhibited a higher accuracy. For example, point B in the first set and point G in the second set had a higher accuracy. However, this phenomenon may have exceptions and requires further investigation with more data.

According to the analysis in Section 2, the first set of fibers (with a core diameter of 600 μm) had 858,050 speckle modes, while the second set of fibers (with a core diameter of 400 μm) had 381,065 speckle modes. The greater the number of speckle modes, the more information is contained in the speckle pattern, and the more features the deep learning model can detect, resulting in more accurate predictions. Therefore, due to its higher number of speckle modes, the first set of fibers overall had higher prediction accuracy than the second set.

The lower accuracy of the second set of fibers can be attributed not only to the reduced number of speckle modes but also to their longer fiber length. A longer fiber length may lead to more interference, and the optical signal attenuates as it propagates through the fiber. If the signal attenuation is too great, it may reduce the contrast of the speckle pattern, making it difficult for the deep learning model to accurately extract features, thus affecting the accuracy.

Despite the reduction in extractable features in the speckle images due to the decrease in core diameter from 600 μm to 400 μm and the increase in fiber length from 65 cm to 100 cm, which exposes a longer section of the fiber to interference, the prediction accuracy only slightly decreased and remained at a very high level overall.

## 5. Conclusions

This study presents a deep learning-based approach for multimode fiber temperature and position sensing using a CNN model to predict temperature and position from speckle images. The model achieved a high accuracy in both tasks across various experimental settings and maintained a high performance in a dual-task framework. The results show that the method efficiently predicts temperature changes and heating positions in fibers of different diameters, highlighting its practicality and potential for broader use. Future work can focus on optimizing the model’s architecture and improving the image processing techniques to further enhance its accuracy and stability in real-world applications.

## Figures and Tables

**Figure 1 sensors-25-02811-f001:**
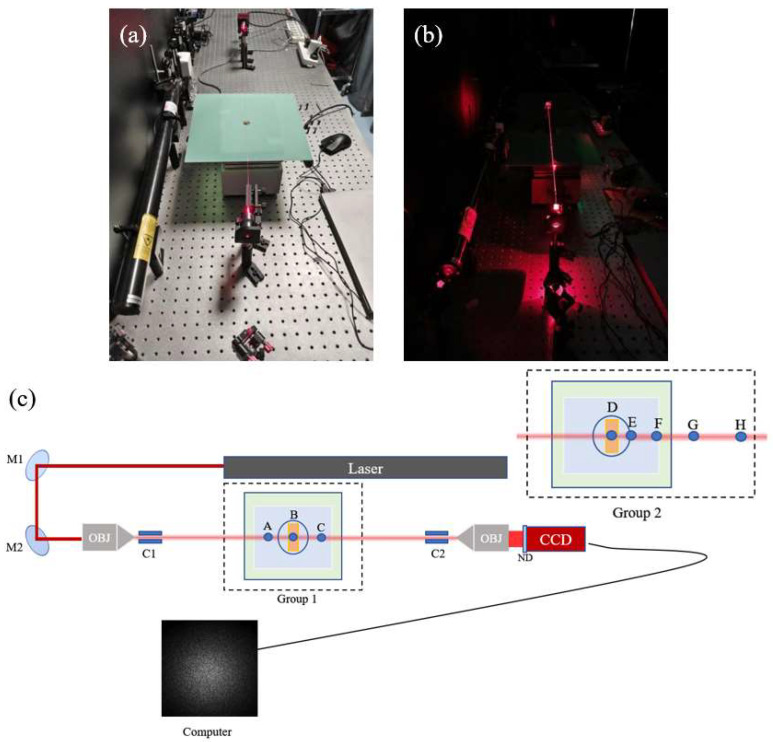
Experimental setup diagram: (**a**) shows the experimental setup under illuminated conditions, (**b**) depicts the setup for speckle acquisition in the dark, and (**c**) presents a top-view schematic of the apparatus. The first experimental configuration is labeled “Group 1” in a dashed box, while the second configuration is labeled “Group 2”, with all other components remaining the same.

**Figure 2 sensors-25-02811-f002:**
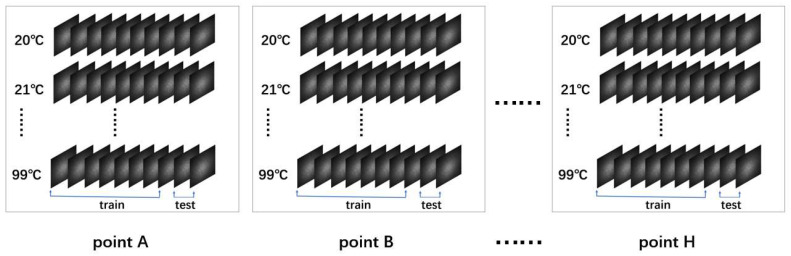
Establishment of dataset.

**Figure 3 sensors-25-02811-f003:**
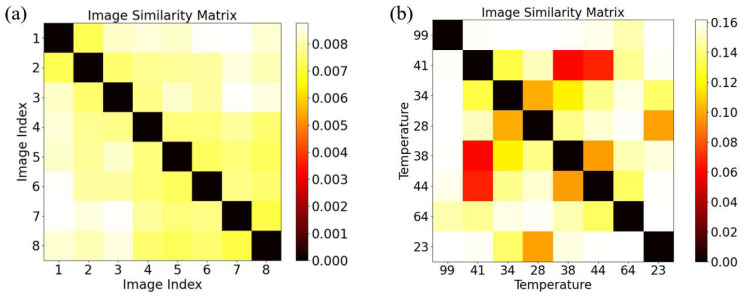
Similarity matrix of point: (**a**) displays the similarity matrix based on eight training images at 20 °C; (**b**) shows a similarity matrix from speckle images of eight randomly chosen temperature points from the training set (out of 80 datasets).

**Figure 4 sensors-25-02811-f004:**
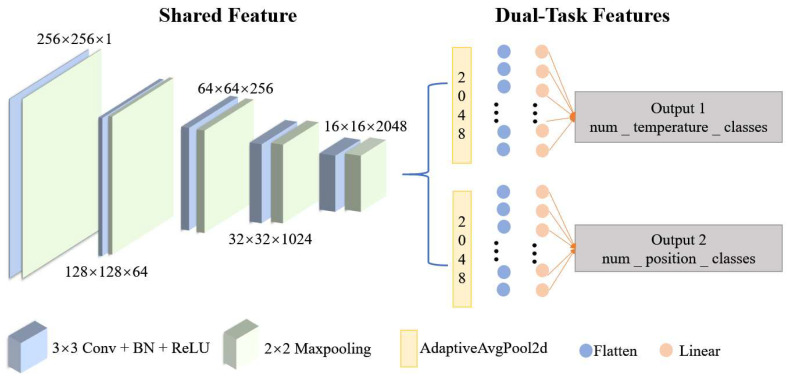
Dual-task learning framework.

**Figure 5 sensors-25-02811-f005:**
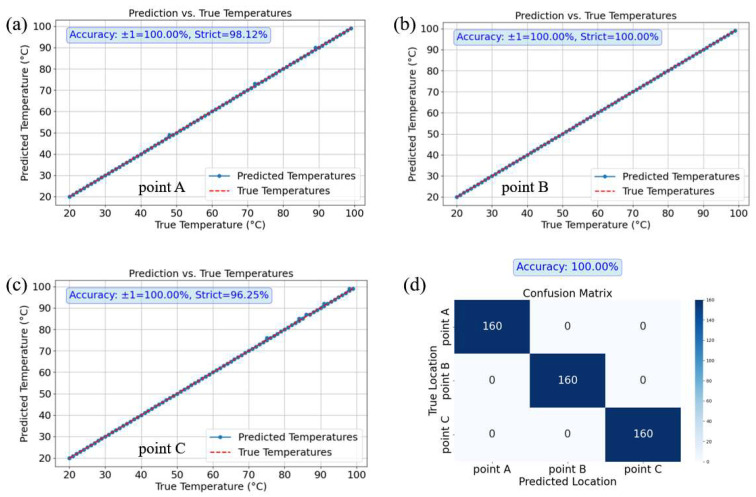
Prediction results of Group 1. (**a**) shows the temperature prediction curve for Point A; (**b**) shows the temperature prediction curve for Point B; (**c**) shows the temperature prediction curve for Point C. All comparing the predicted temperatures with the actual temperatures. (**d**) presents the confusion matrix for location prediction.

**Figure 6 sensors-25-02811-f006:**
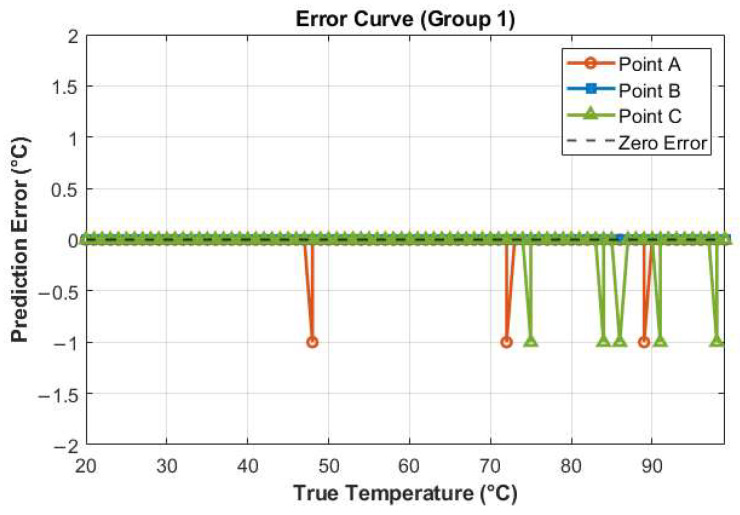
Temperature error curve (Group 1).

**Figure 7 sensors-25-02811-f007:**
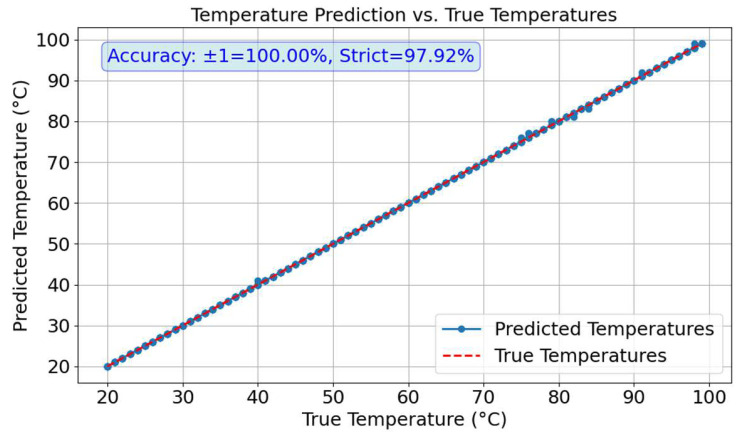
Temperature curves in dual-task prediction.

**Figure 8 sensors-25-02811-f008:**
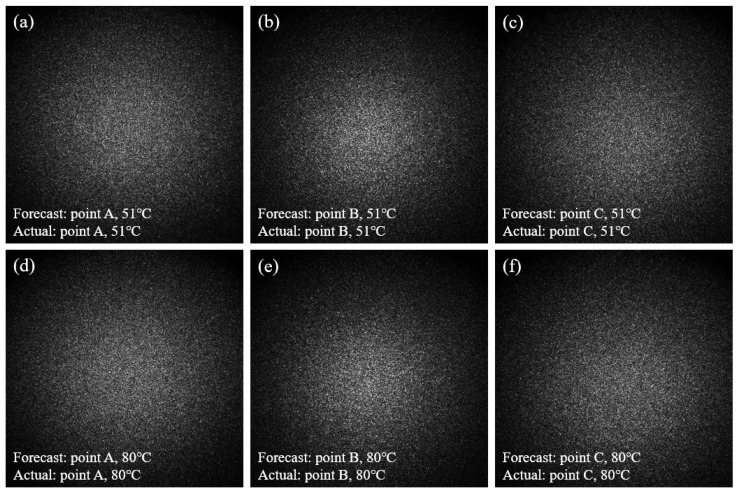
Speckle images and their temperature and position prediction results for Group 1. (**a**) shows that the predicted location of the speckle pattern is Point A, with a predicted temperature of 51 °C; (**b**) shows that the predicted location of the speckle pattern is Point B, with a predicted temperature of 51 °C; (**c**) shows that the predicted location of the speckle pattern is Point C, with a predicted temperature of 51 °C; (**d**) shows that the predicted location of the speckle pattern is Point A, with a predicted temperature of 80 °C; (**e**) shows that the predicted location of the speckle pattern is Point B, with a predicted temperature of 80 °C; (**f**) shows that the predicted location of the speckle pattern is Point C, with a predicted temperature of 80 °C.

**Figure 9 sensors-25-02811-f009:**
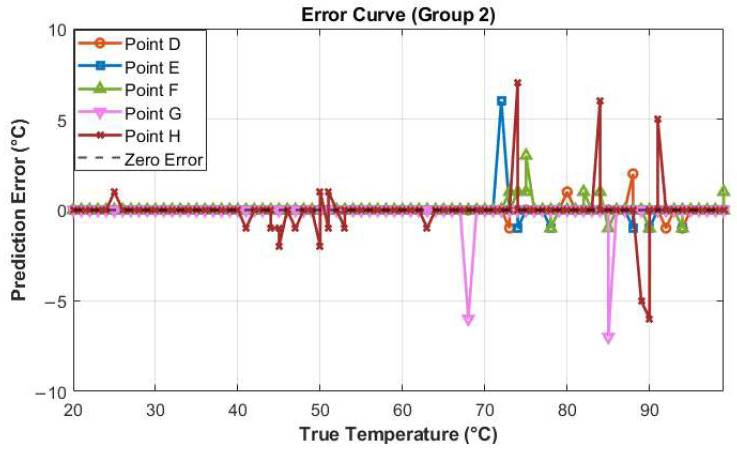
Temperature error curve (Group 2).

**Figure 10 sensors-25-02811-f010:**
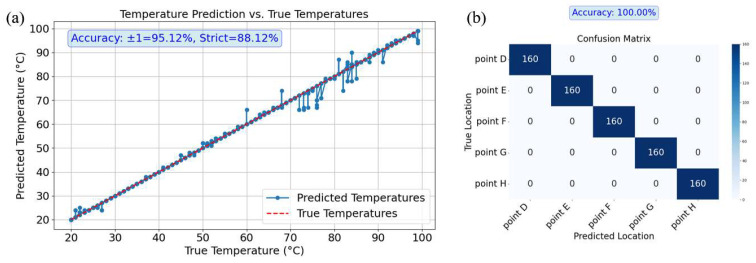
Results of simultaneous temperature and position prediction for Group 2. (**a**) is the temperature prediction curve; (**b**) is the location prediction confusion matrix.

**Figure 11 sensors-25-02811-f011:**
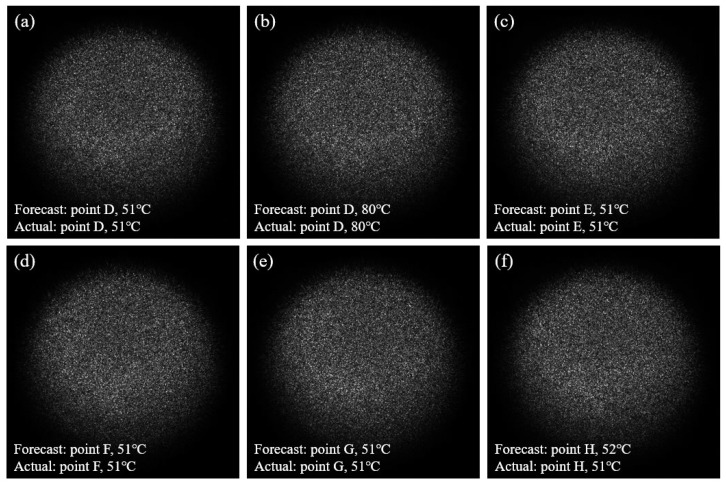
Speckle images and their temperature and position prediction results for Group 2. (**a**) shows that the predicted location of the speckle pattern is Point D, with a predicted temperature of 51 °C, consistent with the actual value; (**b**) shows that the predicted location of the speckle pattern is Point D, with a predicted temperature of 80 °C, consistent with the actual value; (**c**) shows that the predicted location of the speckle pattern is Point E, with a predicted temperature of 51 °C, consistent with the actual value; (**d**) shows that the predicted location of the speckle pattern is Point F, with a predicted temperature of 51 °C, consistent with the actual value; (**e**) shows that the predicted location of the speckle pattern is Point G, with a predicted temperature of 51 °C, consistent with the actual value; (**f**) shows that the predicted location of the speckle pattern is Point H, with a predicted temperature of 52 °C, while the actual temperature is 51 °C, resulting in an error of 1 °C.

**Table 1 sensors-25-02811-t001:** Temperature prediction accuracy for each point in the second group of experiments.

Location	D	E	F	G	H
Tolerance Accuracy	99.38%	98.75%	99.38%	98.12%	95.00%
Strict Accuracy	95.62%	95.62%	91.38%	98.12%	86.25%

## Data Availability

The data presented in this study are available on request from the corresponding author.

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
