# Peer review of "Deep Learning-Based Multimode Fiber Distributed Temperature Sensing"

_sensors, 2025, doi:10.3390/s25092811_

Round 1

Reviewer 1 Report

Comments and Suggestions for Authors
  1. Please provide the temperature range and accuracy of the heating device.
  2. The results shown in Figures 5 (b)show the strict accuracy 100%, Please double check.
  3. It is suggested that the authors test the effectiveness and accuracy of this method within a higher temperature range for applications in hazardous environments.

Reviewer 2 Report

Comments and Suggestions for Authors

In the manuscript entitled “Deep Learning-Based Multimode Fiber Distributed Temperature Sensing”, convolutional neural networks were introduced into fiber-optic temperature sensing to determine temperature and heating location simultaneously. Deep learning was applied to enhance the accuracy of temperature sensing and heating location determination. There are several comments and suggestions as follows:

  1. By using deep learning to distinguish the features of speckle patterns, the temperature and heating location can be determined. To make the article more readable, it is suggested to explain the physical principle of deep learning-assisted fiber-optic temperature sensing. Why does the speckle pattern could reflect the temperature and location?
  2. The camera had an integration time of 8 s to enhance speckle clarity and contrast. It seems too slow for temperature sensing. Could the authors give some discussion on it and how to improve it?
  3. The prediction accuracy of temperature and heating location can reach 100%. The total sample numbers should be indicated. In addition, the accuracy of type two fiber was poorer than that of type one. Could the authors give some explanations on it?
  4. Could the authors discuss the sensing distance of the deep learning-assisted fiber distributed temperature sensing technology?
Comments on the Quality of English Language

The English could be improved to more clearly express the research.

Reviewer 3 Report

Comments and Suggestions for Authors

The manuscript (sensors-3606875) is devoted to the study deep learning-based multimode fiber distributed temperature sensing

1. Typically, optical fibers operate in environments with a temperature range from -40 to +60.
Why is the entire temperature range not covered?

2. You are used industrial CCD camera - please specify the model and her parameters.

3. Which C1 and C2 fiber clips did you use ?

4. Does the accuracy of temperature prediction somehow depend on the optical fiber cladding?

5. What is the maximum possible sensitivity of speckle structures to temperature changes and position determination in your model?

Reviewer 4 Report

Comments and Suggestions for Authors

This work presents alternative fiber optic sensing method for temperature estimation, based on developed dual-output Convolutional Neural Networks (CNN) application for analysis of speckle pattern, measured at the end of large core multimode optical fiber (MMF), excited by laser source. Authors verified proposed solution by the series of experimental tests, that demonstrated strong agreement between predicted and true temperature (average error is less 5%). 
The manuscript is well prepared and organized. The paper corresponds to the journal scope. It is suitable for publication in Sensors after minor revision / improvements / corrections / suggestions and answering on following questions / comments:
1.    There are no any details of utilized MMFs: core diameter, refractive index profile.  Are they silica-silica (core-cladding) or silica-polymer optical fibers? What are 400 um and 600 um outer diameters – are they coating or cladding diameters?
2.    Fig. 5 and 6 demonstrate great agreement between predicted and set temperature with extremely low error, and, therefore, linear dependencies are very close to each other. It is proposed to represent (just for one example) the same linear dependence for absolute (difference) or relative (in %) error.
3.    How long did it take to reach “steady state”, when the temperature changed by 1oC  over tested range 20…99oC?
4.    During sensor system operation in real conditions, should speckle measurements be performed continuously in a real time mode? How often (frequency or time interval)? 
5.    Has it been assessed how quickly the developed sensing system can respond to temperature changes?
6.    Authors tested short (1 m and less) lengths of MMFs to develop and verify temperature estimation and heating point localization method. Is it possible to apply proposed solution for long distance remote fiber optic temperature sensor applications? How sensitive is the developed system to external mechanical influences like tension, stress, torsion, vibrations etc.?
7.    General comments: the abbreviation CNN should also be deciphered in the Introduction (#58).
